

# Observation-inferred resilience loss of the Amazon rain forest possibly due to internal climate variability

Raphael Grodofzig[1,2], Martin Renoult[2,3], and Thorsten Mauritsen[1,2]

[1]Department of Meteorology, Stockholm University, Stockholm, Sweden
[2]Bolin Centre for Climate Research, Stockholm University, Stockholm, Sweden
[3]Department of Geological Sciences, Bolin Centre for Climate Research, Stockholm University, Stockholm, Sweden

**Correspondence:** Raphael Grodofzig (raphael.grodofzig@misu.su.se)

**Abstract.** Recent observation-based studies suggest that the Amazon rain forest has lost substantial resilience since 1990, indicating that the forest might undergo a critical transition in the near future due to global warming and deforestation. The idea is to use trends in lag-1 auto-correlation of leaf density as an early warning signal of an imminent critical threshold for rain forest dieback. Here we test whether the observed change in auto-correlations could arise from internal variability

by using historical and control simulations of nine sixth-generation Earth system model ensembles (Phase 6 of the Coupled Model Intercomparison Project, CMIP6). We quantify trends in leaf area index auto-correlation from both models and satellite observed vegetation optical depth from 1990 to 2017. Four models reproduce the observed trend with at least one historical realization, whereby the observations lie at the upper limit of model variability. Three out of these four models exhibit similar behavior in control runs, suggesting that historical forcing is not necessary for simulating the observed trends. Furthermore,

we do not observe a critical transition in any future runs under the strongest greenhouse gas emission scenario (SSP5-8.5) until 2100 in the four models that best reproduce the past observed trends. Hence, the currently observed trends could be caused simply by internal variability, and, unless the data records are extended, have limited applicability as an early warning signal. Our results suggest that the current rapid decline in Amazon rain forest coverage is mainly caused by local actors.

## 1 Introduction

The resilience level of the Amazon rain forest to external stresses, such as global warming and deforestation, is subject to ongoing debate (Feldpausch et al., 2016; Boers et al., 2017; Boulton et al., 2022). Since the 1970s the Amazon forest has lost about 20 percent of its coverage (Simmons et al., 2019) and the net carbon uptake of the formerly persistent carbon sink has been declining over the last four decades due to intensification of the dry season and elevated deforestation (Gatti et al., 2021). The rapid development has raised warnings that the Amazon forest is approaching a critical threshold, beyond which

irreversible damage is unavoidable (Brando et al., 2014; Boers et al., 2017; Boulton et al., 2022; Parry et al., 2022; Doughty et al., 2023).

    Model studies, however, differ widely in their results and the inter-model spread of vegetation responses in future projections remains high with moderate forest resilience this century, but a higher risk of sporadic rain forest loss past 2100 (Huntingford et al., 2013; Boulton et al., 2017; Chai et al., 2021). Notably, Parry et al. (2022) found localized rain forest dieback using an



abrupt-shift-detection algorithm in five out of seven CMIP6 models they investigated in simulations wherein $CO_2$ is increased by 1 percent per year until quadrupled after 140 years. All in all, though, climate models do not predict an imminent and complete collapse of the Amazon rain forest. However, it has been reported that climate models underestimate vegetation related feedbacks (Richardson et al., 2013; Green et al., 2017; Forkel et al., 2019).

Observation-based studies of the recent historical record convey a more alarming picture. Tao et al. (2022) reported the capacity of undamaged rain forests to withstand future droughts to be limited, especially in the Amazon. Analyzing remotely sensed vegetation data, Boulton et al. (2022) supported this idea and found that more than three quarters of the Amazon rain forest has been losing resilience since the 2000s, especially in regions of less rainfall and in proximity to regions of human activity. They present evidence for an imminent tipping point of the rain forest in the near future.

Such a tipping point may be initiated by a major tree loss from fires or deforestation, or climate change (Cox et al., 2008; Brando et al., 2014). Land cover transitions, such as forest-to-crop or forest-to-pasture, decrease the net surface radiation and latent heat flux, while increasing the sensible heat flux, resulting in warming of the land surface (Silvério et al., 2015). Reducing the vegetation density by deforestation is associated with enhanced precipitation run-off and reduced evapotranspiration. Hence, both deforestation and forest degradation by droughts weaken the moisture transport by recycling, which is mainly directed westwards over the Amazonian basin along the prevalent wind direction (Salati et al., 1979). This causes reduced precipitation downwind, and degraded forest health in a positive feedback loop.

Negative feedbacks and stabilising effects may also exist. For instance vegetation responds positively to increasing levels of $CO_2$ (Kolby Smith et al., 2016), provided sufficient water and nutrients are available, something which can be observed to happen in most parts of the world, including the Amazon basin (Zhu et al., 2016). Another possible mechanism could be convective clouds, that actively shift precipitation from wet to dry regions: the temperature gradient, arising from evaporative cooling in wet regions while warming dry regions, activates a low-level breeze that transports moisture to the dry areas (Hohenegger and Stevens, 2018). This, in extension would be a negative feedback in the Amazon in that the atmosphere acts to moisten dry regions. Since the Amazon rain forest has existed for at least thousands (Malhi et al., 2004), or even millions of years (Maslin et al., 2005), the rain forest must have been dominated by negative feedback in the past.

Tipping points are typically accompanied by a regime shift from a stable state where negative feedback mechanisms dominate, to a marginally stable state with transition to a net positive feedback parameter. Several statistical metrics, known as early warning signals, have been proposed to predict a regime shift (Scheffer et al., 2009; Lenton et al., 2012). Most commonly, they quantify the recovery rate of the system to small perturbations. The resilience of a system is subsequently defined as the ability to recover from those disturbances. As the stability of the system decreases, it recovers slower when stochastically forced. This phenomenon, known as critical slowing down (CSD), can be detected by an increase in lag-1 autocorrelation (AR(1)) of time series representing the dynamics of the system. Increasing AR(1) has been widely used as an early warning signal on Earth's dynamical systems such as the western Greenland ice sheet (Boers and Rypdal, 2021) or the Amazonian rain forest (Boulton et al., 2022). However, the increase in AR(1) has been shown to occur likewise for other physical reasons, as well as not occur prior critical transitions, for instance when the rate of forcing is higher than the intrinsic response time scale for CSD (Boulton et al., 2013).





Here, we compare the observational record to nine large CMIP6 historical model ensembles and control simulations, quantifying model forest resilience between 1990 and 2014 using the same method as in Boulton et al. (2022). By analysing large model ensembles, we can test whether internal variability in auto-correlation could be the source of the observations-inferred resilience loss.

## 2  Methods

### 2.1  Data

We use the Amazon basin as our region of study, taken to be the domain defined as by RAISG - Amazon Network of Georeferenced Socio-Environmental Information, accessed March 2023. The observational data is provided by the Vegetation Optical Depth Climate Archive (VODCA) (Moesinger et al., 2020), which is available in a $0.25° \times 0.25°$ resolution in daily frequency for the period July 1987 to June 2017. The passive or active satellite observations capture the attenuation of microwave radi-

ation by vegetation, which is known as the vegetation optical depth (VOD). The attenuation depends on various factors like density, type and water content of the vegetation and the wavelength range of the sensor (Owe et al., 2008). Shorter wavelengths are more sensitive to upper leaf canopy than longer wavelengths, since they experience higher attenuation by vegetation. We chose the lowest wavelength product available (Ku-band, $\sim 19\,\mathrm{GHz}$) for the period January 1990 to December 2017, following the study by Boulton et al. (2022). The monthly means of the VODCA product are interpolated to $1° \times 1°$ for better comparing

to models, although this did not substantially affect the observed trend (see Fig. A1).

To assess internal variability, CMIP6 model ensembles with at least seven historical runs available and interactive leaf area index (LAI) are included in the study (Table 1). The non-dimensional LAI is defined as the total area of leaves per unit surface area. Although LAI and VOD are not identical variables, but physically closely related, and changes in both variables are strongly correlated, such that they can be considered good proxies for forest health and resilience (Moesinger et al., 2020).

The model LAI output is evaluated in the period January 1990 to December 2014, as the historical experiments are only available until this point in time. Additionally, we use Shared Socioeconomic Pathway 5 (SSP5-8.5) and pre-industrial control simulations of 500 or 1000 years in length that we cut in windows of 25 years, corresponding to the length of the historical period. That way, we create a control ensemble of 20 or 40 members. Control simulations have the advantage that the internal variability of the model can be directly assessed since no external forcing is present. The land use and land cover change

(LUCC) is determined by using the historical land use harmonization dataset LUH2 v2h (Hurtt et al., 2020), that is available as annual values on a $0.25° \times 0.25°$ spatial resolution and utilized to force the land components of CMIP6 models.

### 2.2  Resilience indicator AR(1)

The random variability of a signal contains information about the recovery rate from stochastic perturbations, such that we can separate trend, seasonality and residual of the signal using seasonal-trend decomposition (STL) by Loess (Cleveland et al.,

1990). Assuming the seasonality is constant in time, we choose the STL input parameters trend = 19, season = 13 and period



**Table 1.** CMIP6 model ensembles used within this study.

| Model | Land surface model | Nominal resolution | Number of hist. runs | Reference |
|---|---|---|---|---|
| ACCESS-ESM1-5 | CABLE2.4 | 250 km | 40 | Ziehn et al. (2020) |
| IPSL-CM6A | ORCHIDEE v2 | 250 km | 33 | Boucher et al. (2020) |
| MPI-ESM1-2-LR | JSBACH 3.2 | 250 km | 30 | Mauritsen et al. (2019) |
| MIROC-ES2L | MATSIRO6.0+VISIT-e v1 | 500 km | 30 | Hajima et al. (2020) |
| CanESM5 | CLASS3.6-CTEM1.2 | 500 km | 25 | Swart et al. (2019) |
| MPI-ESM1-2-HR | JSBACH 3.2 | 100 km | 10 | Mauritsen et al. (2019) |
| INM-CM5-0 | INM-LND1 | 100 km | 10 | Volodin and Gritsun (2018) |
| CESM2 | CLM5 | 100 km | 9 | Danabasoglu et al. (2020) |
| EC-EARTH3-Veg | HTESSEL | 100 km | 7 | Döscher et al. (2022) |

= 12 months, corresponding to choices made in previous work (Boulton et al., 2022). Nevertheless, slightly altering the STL parameters has no relevant influence on the presented results. The residual component of each grid cell can then be used to quantify the short-term responses of the forest by calculating the AR(1) on a sliding window of 5 years. We find the AR(1) time series by using an ordinary least-squares fitting method for the autoregressive model (Eq. 1), where $\epsilon_t$ represents white

noise of the model with zero mean and constant variance $\sigma_\epsilon^2$, $X_t$ is the time series in each grid point and $\varphi$ the autoregressive coefficient, also denoted AR(1).

$$X_t = \varphi X_{t-1} + \epsilon_t \tag{1}$$

We then quantify the trend of this AR(1) time series by Kendall's rank correlation $\tau_K$ that measures how well two data vectors agree on their ranks (Kendall, 1938). Choosing one vector to be time, a Kendall's $\tau_K = 1$ indicates a strictly increasing

AR(1) trend, $\tau_K = -1$ a strictly decreasing and $\tau_K = 0$ no trend. The significance $p$ of this statistic is computed by randomly generating phase surrogates of the time series' Fourier transform under constant variance and serial correlation (Dakos et al., 2008).

For comparison between observations and models, we calculate both the $\tau_K$ of the spatially averaged AR(1) series and for individual grid cells. The spatial distributions of $\tau_K$ are tested for similarity between observations and model using the

non-parametric, two-sample Kolmogorov-Smirnov test (Berger and Zhou, 2014). Its test statistic is given by the maximum difference between two cumulative distribution functions and is computed under the null hypothesis that both samples are drawn from populations that have an arbitrary, yet identical, underlying distribution. The higher the $p$-value of this test, the less likely it is that the samples are drawn from different underlying distributions.



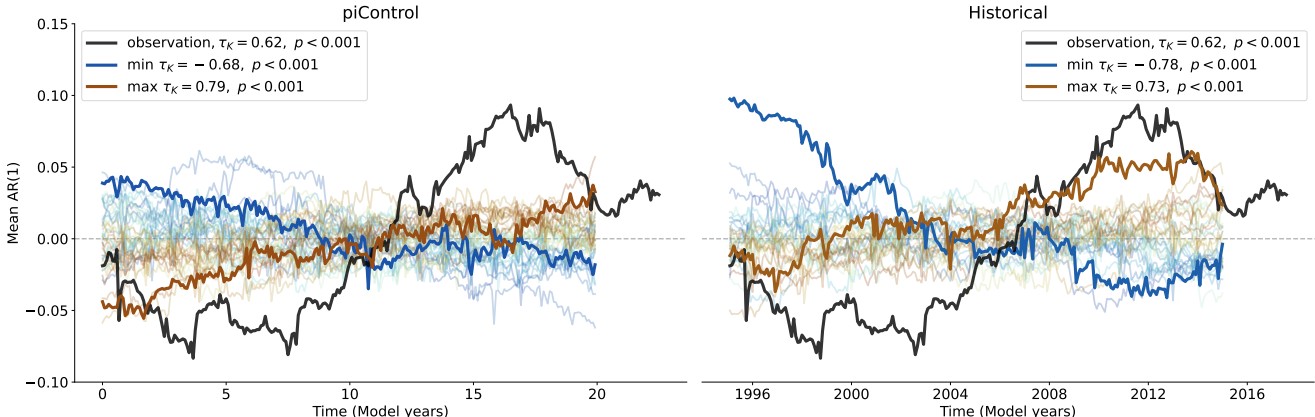

**Figure 1.** Anomaly of the spatially averaged AR(1) series for observations and MPI-ESM1-2-LR piControl (left panel) and historical (right panel) ensemble. The anomaly is computed by subtracting the temporal mean from the time series, and is plotted at the end of the 5 years sliding window. The ensemble member with the largest and smallest $\tau_K$ are highlighted. The color gradient of all ensemble runs corresponds to increasing $\tau_K$.

## 3 Results

The observational record of auto-correlation of VOD from the Amazon basin exhibit variations and trends (Figure 1), but it is impossible based solely on a single data record to figure out whether such variations are caused by external forcing, or it is simply an expression of internal variability. A commonly used method to detect forced changes and events in observational records that contain internal and natural variability is to compare in various ways to multiple climate model runs with different codes and/or starting from different initial conditions (e.g. Hasselmann (1997), Otto (2023)). In particular it has become

common that global climate models are run multiple times with the same historical boundary conditions, but starting from different initial conditions in order to explore their internal variability (Kay et al. (2015), Maher et al. (2021), Table 1). With such large ensembles it is possible to ask whether the observed trend is within the range of variability exhibited by the model, and in extension in the present case whether an increasing trend in auto-correlation constitutes a skillful early warning signal.

      As an example, we display 30 simulations of the historical experiment together with one CMIP6 model, MPI-ESM1-2-LR,

together with the observations (Figure 1, right panel). We see that the observed trend in spatially averaged AR(1) of $\tau_K =$ 0.62 is within the range of trends exhibited by the model ensemble members (-0.76 to 0.72) when calculated over the same period. Therefore, in terms of this model's behavior the observed trend is within the range of variability. To investigate whether global warming or land-use change is affecting the trends we can also inspect equally long duration chunks from the same model's pre-industrial control simulation (Figure 1, left panel). Here we also find trends that encapsulate the observed trend

(-0.68 to 0.79). The shorter term deviations appear slightly muted in this case compared to the historical ensemble, and even if encapsulating the longer term trend the shorter variations are clearly less than those observed, therefore possibly suggesting





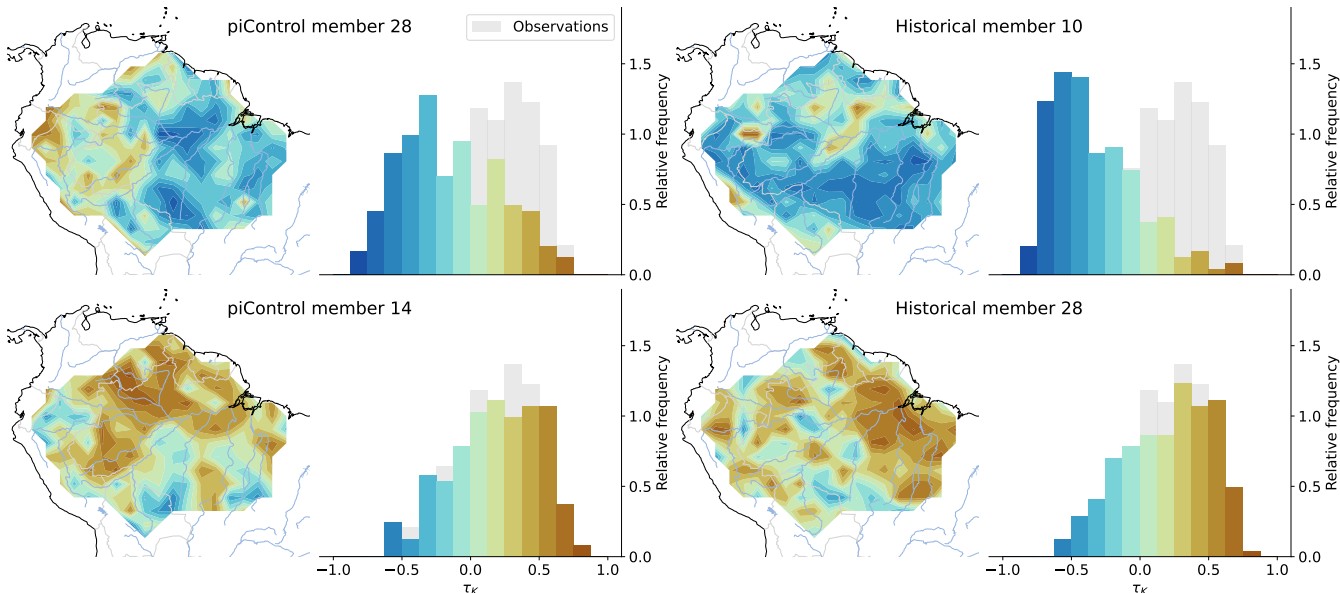

**Figure 2.** Maps and histograms of the $\tau_K$ values of selected members from the piControl (left panels) and historical (right panels) MPI-ESM1-2-LR ensemble. The observed $\tau_K$ distribution is plotted in grey for comparison. We selected the ensemble members that have the highest and lowest $\tau_K$ of the spatially averaged AR(1) series to present the full range of model variability.

an influence from historical forcing. Overall, however, the displayed model ensemble suggests that the forced response of the Amazon rain forest is not needed to generate an increase in AR(1) of similar magnitude to that observed.

The modelled spatial distributions of the trends in AR(1) further support the idea that the origin is internal variability. We can
inspect maps and frequency distributions from the two historical ensemble members and the two chunks from the pre-industrial control in Figure 2 that exhibited the largest trends (Figure 1). All historical ensemble members are shown in Figure A2. We see that positive trends can occur in any part of the Amazon basin, and are not preferentially occurring in the southern parts where most of the land use changes happened (Figure A3). There are, however, cases wherein this pattern occurs (e.g. members 4, 13, 15, 26 and 30). Also noteworthy is that the model does display trend patterns with spatial scales that are substantially larger
than the model resolution such that the underlying causes must be a simulated feature of the model, e.g. large scale weather events.

Not all models are equally fit for the purpose of simulating the Amazon rain forest dynamics in accordance with observations. To investigate this, we apply the Kolmogorov-Smirnov (KS) test (Section 2.2) on each ensemble member to test whether it could have been drawn from the same underlying distribution as the observations. The process is illustrated in Figure 3. For the
MPI-ESM1-2-LR model the test identifies two ensemble members (5 and 28) that are statistically indistinguishable from the observed trend distribution. We carry out the same procedure for all models and find that four of the nine models pass the test (Table 2) with CanESM5, ACCESS-ESM1-5 and MIROC-ES2L showing closer affinity to observations than the MPI-ESM1-2-LR model which we have focused on thus far.







**Figure 3.** Cumulative distribution functions of $\tau_K$ for observations and members of MPI-ESM1-2-LR. The KS-statistic is drawn as a black error bar. Members that are not significantly different from the observational distribution are marked in red. Member 28 shows best agreement with KS $= 0.08$ and $p = 0.22$.

A drawback of applying the KS test method here is that some model ensembles may be too small, and therefore simply by

chance non of the ensemble members pass the KS test, even if the model is capable of producing such a simulation. Indeed, we see that four of the five models that are not passing the KS test have 10 or fewer historical ensemble members. For example MPI-ESM1-2-HR is physically very similar to MPI-ESM1-2-LR, with the main difference of applying a higher resolution resulting in fewer simulated members: 10 instead of 30. Therefore it is plausible that also this model is fit for purpose, but it was not run enough times to demonstrate that. The only exception to this rule is IPSL-CM6A, which with 33 ensemble

members is extremely unlikely to have a plausible representation of Amazon vegetation dynamics.

 

**Table 2.** Observation's percentiles $p_{cml}$ in mean $\tau_K$ distribution, fraction of ensemble members $F_{p>0.05}$ that scored $p > 0.05$ in the KS-test and highest KS-test $p$-value of each ensemble. The higher the $p$-value, the more likely the member resembles the observations. Displayed for all nine historical ensembles and the control ensembles of the four well-agreeing historical ensembles. Well-agreeing ensembles have at least one member that does not significantly differ from the observations according to the KS test.

| Model | historical | | | piControl | | |
|---|---|---|---|---|---|---|
| | $p_{cml}$ of $\overline{\tau}_K$ | KS-test $F_{p>0.05}$ | KS-test highest $p$ (member) | $p_{cml}$ of $\overline{\tau}_K$ | KS-test $F_{p>0.05}$ | KS-test highest $p$ (member) |
| MIROC-ES2L | 0.04 | 6/30 | 0.97 (13) | $< 0.01$ | 1/20 | 0.29 (9) |
| ACCESS-ESM1-5 | 0.02 | 3/40 | 0.56 (13) | 0.01 | 1/20 | 0.21 (12) |
| CanESM5 | $< 0.01$ | 2/25 | 0.75 (24) | 0.02 | 4/40 | 0.84 (20) |
| MPI-ESM1-2-LR | 0.01 | 2/30 | 0.22 (28) | 0.02 | 3/40 | 0.40 (14) |
| MPI-ESM1-2-HR | $< 0.01$ | 0/10 | $< 0.05$ (-) | | | |
| CESM2 | $< 0.01$ | 0/9 | $< 0.05$ (-) | | | |
| EC-EARTH3-Veg | $< 0.01$ | 0/7 | $< 0.05$ (-) | | | |
| IPSL-CM6A | $< 0.01$ | 0/33 | $< 0.05$ (-) | | | |
| INM-CM5-0 | $< 0.01$ | 0/10 | $< 0.05$ (-) | | | |

Inspecting the Amazon mean trend of AR(1) for each of the ensemble members from the four models that passed the KS test we find one model which displays a significant response to historical forcing (MIROC-ES2L, Figure 4). To better show the shift between the pre-industrial control and historical simulations we fit a bounded beta-distribution to the frequency distribution. In all four models and for both simulations the observed trend is on the edge of what is possible. In CanESM5, MPI-ESM1-2-LR and ACCESS-ESM1-5 there is not a substantial difference between the forced and unforced simulations, but MIROC-ES2L shows a marked shift towards higher trends in Amazon basin mean AR(1) when the model is exposed to historical boundary conditions.

We finally test the idea that an increasing AR(1) can be used as an early-warning signal of an eminent abrupt transition in the Amazon rain forest. This was done by inspecting the continuation future projections of the two historical runs with largest positive and negative historical trends from MPI-ESM1-2-LR (Fig. 5). This is the Shared Socioeconomic Pathway 5 with a radiative forcing of about $8.5\,\mathrm{W/m^2}$ at the end of the Century (SSP5-8.5). Under this strong forcing future scenario the model exhibits a slight increasing trend in AR(1), along with a decreasing trend of LAI of on average $-0.05\,\mathrm{m^2m^{-2}}$. Neither of the two extreme runs, nor any other ensemble members exhibit an abrupt decline in LAI. In fact, we do not observe a regime shift in any of the other four models that passed the KS test. The linear deterioration of forest viability can be caused by a variety of factors including limited water availability or progressing land use change. However, the changes induced by future forcing do not facilitate bifurcation-like behavior of the system.



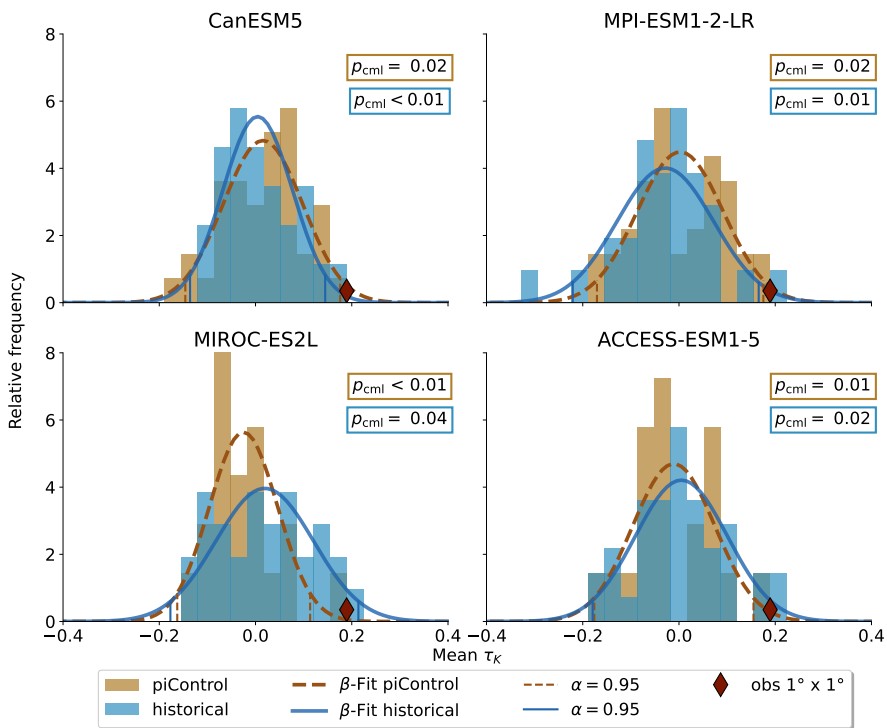

**Figure 4.** Model's mean $\tau_K$ distributions for piControl and historical runs. The histograms are fitted with a $\beta$-distribution on the interval $[-1, 1]$. The observation's percentile within the respective distribution is denoted by $p_{cml}$. Historical forcing makes the occurrence of the observations more likely only in MIROC-ES2L and ACCESS-ESM1-5, while CanESM5 and MPI-ESM1-2 simulate a higher probability in control runs.

## 4  Conclusions

In this study we have tested the idea that trends in the persistence of vegetation density anomalies can be used as an early
warning signal for Amazonian rain forest. This is particularly concerning upon the backdrop of a large observed trend since
1991, suggesting that the forest has undergone a pronounced loss of resilience (Boulton et al. 2020). The trend in anomaly
persistence is quantified through the lag-1 year correlation, AR(1).

To this end, we inspect simulations from nine Earth system model ensembles initialized with different initial conditions in
1850, such that variations within each ensemble is an expression of internal variability simulated by that particular model.
We find that four of the models have ensemble members that are statistically indistinguishable from the observed trend. Four
other models with 10 or fewer realisations did not have a matching realisation, and one model with 33 realisations did clearly
under performs. Of the four well performing model ensembles, three of them showed trends similar to observations also in
their unforced control simulations. These results suggest that the observed trend could simply be an expression of internal
variability, and that longer data records would be needed to show that the opposite is the case.



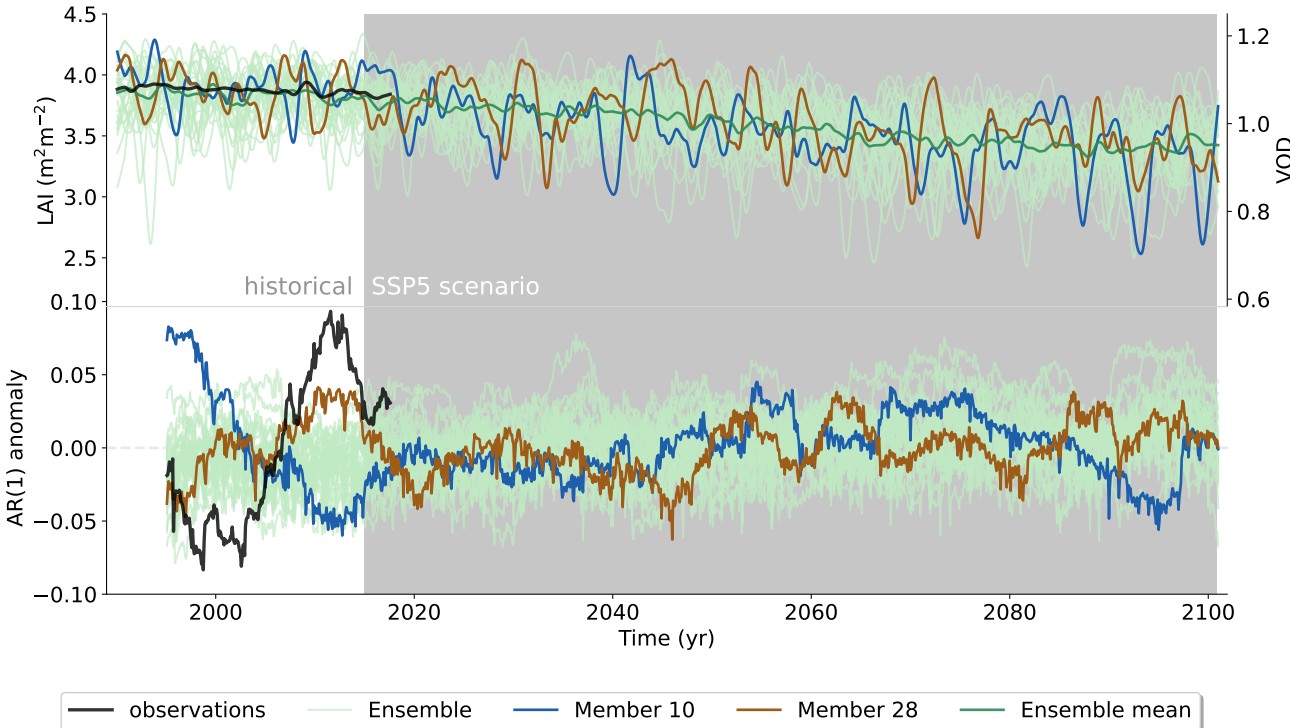

**Figure 5.** Historical and SSP5-8.5 scenario of MPI-ESM1-2-LR LAI and corresponding spatially averaged AR(1) trend compared to the observational VOD record. Upper panel displays the STL trend component of LAI and VOD respectively, lower panel shows the AR(1) series of the STL residual plotted at the end of the 5 years sliding window. Ensemble member 28 is the best-agreeing member with observations in the historical period, member 10 the least agreeing. Note that the dual vertical axes used in the upper panel are scaled to have the same relative range.

This result is further corroborated by the spatial distribution of the increasing trend in AR(1) in the model simulations. Here it is found that ensemble members with substantial positive or negative trends show these in relatively large regions, but not necessarily in those regions with large anthropogenic deforestation. This suggests that such anomalies are associated with large scale weather events.

        We finally check whether trends in AR(1) can be used as an early warning signal by investigating the relationship between
the recent past and the rain forest evolution in future strong warming scenario projections (SSP5-8.5). However, there is no such relationship, and furthermore none of the future simulations exhibit rapid transitions.

        It is worth noting that even if the results presented here suggest that the Amazon rain forest has not lost its resilience, and is unlikely to undergo bifurcations in the future, it does not mean the forest is invulnerable to human caused stresses from global warming, deforestation and fires. On the contrary, the results presented here suggest that the observed rapid decline in rain



forest extent is foremost caused by local actors, and that global warming only plays a minor role. Hence, mitigation strategies to limit future rain forest loss should focus foremost on local stakeholders.

*Data availability.*  The data of the CMIP6 models can be downloaded from ESGF Portal at DKRZ, located at https://esgf-data.dkrz.de/projects/esgf-dkrz/ (last access: 15 May 2023). The observational VOD data is available on https://doi.org/10.5281/zenodo.2575599 (last access: 18 March 2023). Land use and land cover data can be found on https://luh.umd.edu/index.shtml (last access: 21 April 2023). The outlines of the

Amazon forest are taken from RAISG https://www.raisg.org/en/ (last access: 10 March 2023).

*Author contributions.*  RG carried out the analysis and drafted the paper with inputs from MR and TM. All authors contributed to the study.

*Competing interests.*  The contact author has declared that none of the authors has any competing interests.

*Acknowledgements.*  This work was supported through funding from the Swedish Research Council (VR), Grant agreement 2022-03262, the European Research Council (ERC) Grant agreement 770765 and the European Union's Horizon 2020 program Grant agreements 820829

and 101003470. We acknowledge the World Climate Research Program, which is in charge of CMIP, and thank the climate modeling groups (listed in Table 1 in the Methods section) for producing and publishing their model output. We acknowledge equally Moesinger et al. (2020) for creating the global long-term VOD Climate Archive (VODCA) and making their data available.



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



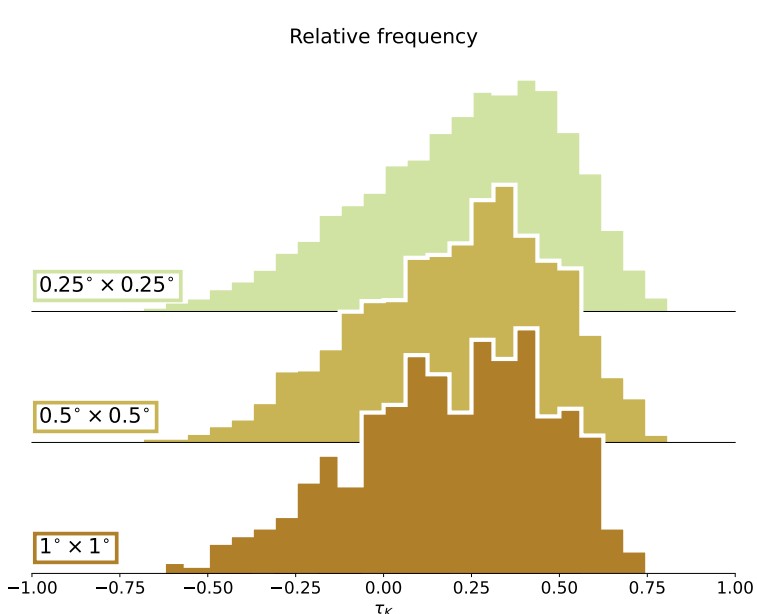

**Figure A1.** Kendall's $\tau_K$ distributions for three different VOD resolutions ($0.25° \times 0.25°$ original VODCA). The means of these histograms are respectively $\bar{\tau}_{K,0.25°} = 0.20$, $\bar{\tau}_{K,0.5°} = 0.20$ and $\bar{\tau}_{K,1°} = 0.19$. We use the $1° \times 1°$ resolution for our analysis since it is the closest to the model resolutions.





**Figure A2.** Maps of $\tau_K$ of MPI-ESM1-2-LR members according to Fig. 3. The spatial mean for each run is indicated, the observational value is $\overline{\tau}_K = 0.21$ (resolution $1° \times 1°$).



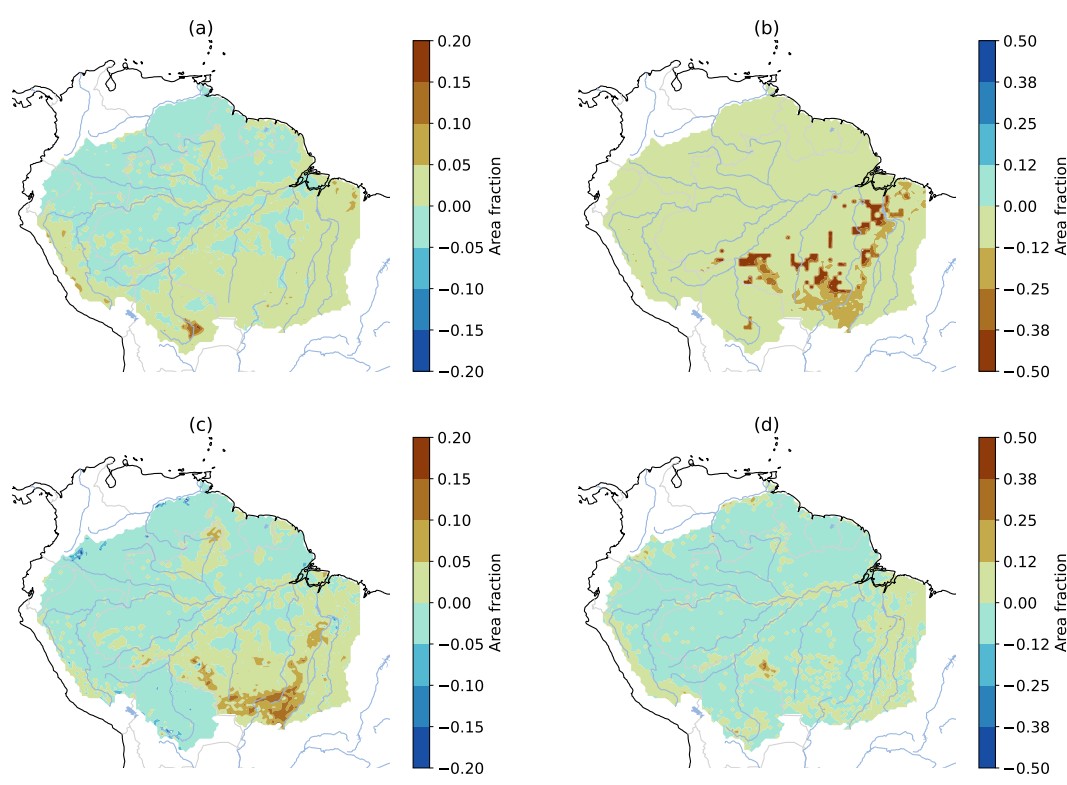

**Figure A3.** Difference between 2014 and 1990 of (a) C3 annual crops (most small-seeded cereal crops), (b) primary forested land, (c) rangeland and (d) pasture. Agricultural activity (a, c and d) increased particularly in the south-eastern regions of the Amazonian, while primary forested land (b) declined in the same areas.





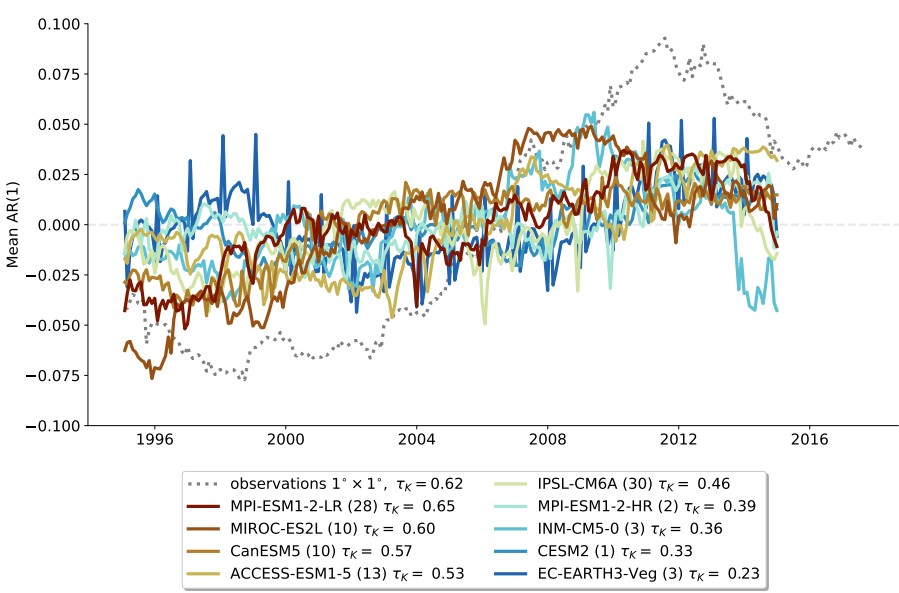

**Figure A4.** Anomaly of the spatially averaged AR(1) series for observations and best agreeing member for each historical ensemble. The anomaly is computed by subtracting the temporal mean from the time series, and is plotted at the end of the 5 years sliding window. The best agreeing member is shown in parentheses behind the model (the member number shown here agrees with the realization number of the CMIP6 variant ID) and chosen according to its trend statistic $\tau_K$.