# Peer review of "Observation-inferred resilience loss of the Amazon rain forest possibly due to internal climate variability"

_EGUsphere, 2023_

## Referee Comment (RC1)

**Review of "Observation-inferred resilience loss of the Amazon rain forest possibly due to internal climate variability"**

The manuscript by Grodofzig et al. studies trends in lag-1 auto-correlation, AR(1), of leaf area index in the Amazon rain forest as simulated by an ensemble of Earth system models. The study complements recent observational evidence by providing a modeling perspective on the statistical significance of estimates. As increased lag-1 auto-correlation is an indicator for the loss of forest resilience, which can serve as an early warning signal for the system to reach a tipping point, the understanding of natural and forced trends in AR(1) is highly relevant. The paper is concise and well-written but I have a few general suggestions that would in my opinion provide the reader with a better understanding of the results and strengthen the scientific significance of the findings.

**General comments**

- Tom my understanding, the authors reproduce the methodology from Boulton et al. (2022). Nevertheless, the observational trends plotted in this paper and in Boulton et al. (2022) look markedly different. The trend in Boulton et al. (2022) is nearly monotonic from 2003 to 2016 ($\tau = 0.913$) whereas the AR(1) trend in the manuscript under review shows a decrease from $\sim 2012$ and 2016. Together with the inclusion of the period prior to 2003, this seems to contribute to a smaller $\tau$ ($\tau = 0.62$) compared to Boulton et al. (2022). I ask the authors to clarify why the observational curve in the current manuscript is different from the one in Boulton et al. (2022). In addition, it would strengthen the manuscript strongly if the authors would additionally present results for shorter periods (e.g., 2003-2014) given that the results in Boulton et al. (2022) are for the period 2003 - 2016.

- The authors put their work in the context of attribution studies for weather and climate extremes. Using models in attribution studies is only justified if the models adequately simulate the studied phenomenon. I understand that the observation period of the AR(1) time series is too short to evaluate the models and assess the significance of recent trends. Nevertheless, it does not become clear if the models are adequately simulating Amazon rain forest dynamics. A skillful simulation of leaf area index across the Amazon rain forest would substantially increase the confidence in the simulated AR(1) trends. Therefore, I ask the authors to (1) provide a more extensive description of the used models (how complex are they compared to state-of-the-art vegetation models? do all of them simulate vegetation dynamically?), and (2) either refer to previous studies in which the ability of the models to simulate the Amazon rain forest is analyzed or include some analysis, e.g., on the similarity of the simulated mean state and spatial patterns with observations. In addition, the authors do only present anomaly time series from the time mean AR(1) coefficients. Are the temporal mean AR(1) coefficients comparable to the time mean AR(1) coefficients of the observations (in the spatial mean and the spatial patterns)?

- The absence of a more extended description of the models also precludes the interpretation of differences between models. In particular, it would be interesting to explore potential reasons for why one model shows a significant difference between the control simulations and the historical period whereas the other models do not show such a change.

- $\tau$ measures the monotonicity of AR(1) trends but it does not quantify the rate of increases/decreases. It would be very interesting to also compare the simulated and observed rate of the increase to assess how exceptional the observed 2003 - 2016 trend is.

**Specific comments:**

- l. 13: After reading the manuscript, this strong statement does not seem to be supported by the presented evidence. Where in the manuscript is the role of local actors studied?

- l. 57: Can you give references or examples for increasing AR(1) due to other physical reasons?

- Sect. 2.1: The description of the used simulations is very short. More information on the complexity of the employed models and similarities/differences between them would be very helpful. In particular, do they all simulate dynamic vegetation (variable PFT coverage frequencies)? If not, are there systematic differences between models with and without dynamic vegetation?

- l. 78: Can you expand on why VOD and LAI are physically closely related?

- l. 78-79: While the manuscript states that LAI and VOD are strongly correlated, this seems to be not the case in the Amazon catchment area (Sect. 4.4 and Fig. 11 in Moesinger et al., 2020). This absence of a strong positive correlation in the Amazon rain forest should be stated explicitly and the use of LAI in models and VOD in observations should be justified in light of this weak correlation.

- l. 104-105: This sentence might be misleading. To my understanding, the employed Kolmogorov-Smirnov test does not explicitly test for similarity of the spatial structure (i.e., spatial correlations) but only compares the distribution of the $\tau$ values across all considered grid boxes. For example, if all the $\tau$ values would be randomly reshuffled (thereby loosing the spatial information), it would not change the result of the KS test.

- l. 119: One 'together' too much

- l. 137-143: What are the implications of the fact that the KS-test is only passed for so few ensemble members?

- l. 149-150: It is not clear to me if such a strong differentiation between IPSL and models, that pass the test for 1 or 2 members, is justified. Can you quantify this difference statistically?

- Table 2: Please explain in the caption how $p_{cml}$ is computed

- Fig. 3: It would be insightful to also plot the KS tests for the other models, for example in a supplement. As a major novelty of the paper is the use of multiple models, it does not seem justified to focus on MPI-ESM in several figures and not give equal exposure to all models (or at least the four models with 20 or more ensemble members).

- l. 152: Can you explain how the significance is computed here?

- l. 159-167: Why is only MPI-ESM1-2-LR used here and not the other ESMs with plausible ensemble members? Table 2 suggests to me that MIROC-ES2L should be given a larger role in the analysis compared to MPI-ESM since it possesses the most ensemble members that pass the KS-test under historical forcing.

- l. 165-166: Is the land use change enforced by the SSP scenario? If yes, it should be easy and insightful to quantify its role.

- l. 171 Boulton et al. (2022) instead of Boulton et al. 2020?

- l. 174: "are" instead of "is"

- l. 178: "underperform" instead of "under performs"

- l. 178-179: While similar trends to the observations are found in a few ensemble members, the conclusions should emphasize stronger how rarely they occur in the ensembles. For example, the observation's percentiles are at most 0.02 for all piControl ensembles (Table 2) and at most 0.02 for all historical ensembles except for MIROC-ES2L (which is also the only model with a significant

change between PI and historical). While it is true that the results suggest that the trend 'could' occur due to internal variability, the likelihood of this occurrence should be the more important quantity, as it is common in other attribution studies.

- l. 182-183: Can you expand on why the results suggest that "large scale weather events" are responsible for the anomalies?

- l. 189-191: The presented results only analyze trends in AR(1), interpreted as an indicator of rain forest resilience. Where in the study are the reasons for the decline in rain forest extent studied and where is the role of local actors analyzed?

**References**

Boulton, C. A., Lenton, T. M., and Boers, N.: Pronounced loss of Amazon rainforest resilience since the early 2000s, Nat. Clim. Chang., 12, 271–278, https://doi.org/10.1038/s41558-022-01287-8, 2022.

Moesinger, L., Dorigo, W., De Jeu, R., Van Der Schalie, R., Scanlon, T., Teubner, I., and Forkel, M.: The global long-term microwave Vegetation Optical Depth Climate Archive (VODCA), Earth Syst. Sci. Data, 12, 177–196, https://doi.org/10.5194/essd-12-177-2020, 2020.

---

## Author Comment (AC1)

**Response to review comments 1**

We thank reviewer 1 for their detailed comments on the manuscript "Observation-inferred resilience loss of the Amazon rain forest possibly due to internal climate variability". In this document we will discuss and respond in detail to the submitted comments.

**General comments**

- *To my understanding, the authors reproduce the methodology from Boulton et al. (2022). Nevertheless, the observational trends plotted in this paper and in Boulton et al. (2022) look markedly different. The trend in Boulton et al. (2022) is nearly monotonic from 2003 to 2016 (tau = 0.913) whereas the AR(1) trend in the manuscript under review shows a decrease from ~ 2012 and 2016. Together with the inclusion of the period prior to 2003, this seems to contribute to a smaller (tau = 0.62) compared to Boulton et al. (2022). I ask the authors to clarify why the observational curve in the current manuscript is different from the one in Boulton et al. (2022). In addition, it would strengthen the manuscript strongly if the authors would additionally present results for shorter periods (e.g., 2003-2014) given that the results in Boulton et al. (2022) are for the period 2003 - 2016.*
  The observational curve presented in our manuscript matches the curve presented by Boulton et al. (2022) in the supplementary material (Figure S5). In this case, the record from 1995 – 2017 shows even a greater AR(1) increase (tau = 0.646) compared to the period
  2003 – 2017 (tau = 0.562). For that reason, and to include the longest observational record as possible in order to capture decadal variability, we decided to include the full available record. The difference of the reported tau by Boulton et al. (2022) of tau = 0.646 and our
  tau = 0.62 can be explained by the grid cell selection procedure (see Boulton et al. (2022) methods). Further, the difference between our reported observational curve and the main text curve by Boulton et al. (2022) originates from the seasonal-trend-decomposition of the signal. As reported, we use the input parameters trend = 19, season = 13 and period = 12 months with the Python statsmodels STL function, gaining the same result as Boulton et al. (2022) in the supplementary material. Robustness tests for the choice of parameters were done by the same authors.

- *The authors put their work in the context of attribution studies for weather and climate extremes. Using models in attribution studies is only justified if the models adequately simulate the studied phenomenon. I understand that the observation period of the AR(1) time series is too short to evaluate the models and assess the significance of recent trends. Nevertheless, it does not become clear if the models are adequately simulating Amazon rain forest dynamics. A skillful simulation of leaf area index across the Amazon rain forest would substantially increase the confidence in the simulated AR(1) trends. Therefore, I ask the authors to (1) provide a more extensive description of the used models (how complex are they compared to state-of-the-art vegetation models? do all of them simulate vegetation dynamically?), and (2) either refer to previous studies in which the ability of the models to simulate the Amazon rain forest is analyzed or include some analysis, e.g., on the similarity of the simulated mean state and spatial patterns with observations. In addition, the authors do only present anomaly time series from the time mean AR(1) coefficients. Are the temporal mean AR(1) coefficients comparable to the time mean AR(1) coefficients of the observations (in the spatial mean and the spatial patterns)?*

(1) Information on dynamic plant functional type coverage is added to Table 1. Details on land surface model components are added in line 85 – 94.

(2) Reference of estimation of LAI representation in models is added. Changes made to line 95 – 97.

We only present AR(1) anomalies because only the change of AR(1) is of interest analysing forest resilience.

- *The absence of a more extended description of the models also precludes the interpretation of differences between models. In particular, it would be interesting to explore potential reasons for why one model shows a significant difference between the control simulations and the historical period whereas the other models do not show such a change.*
As our study is based on a multi-model approach, it is beyond its scope to understand in detail the behaviour of single models.

- *tau measures the monotonicity of AR(1) trends but it does not quantify the rate of increases/decreases. It would be very interesting to also compare the simulated and observed rate of the increase to assess how exceptional the observed 2003 - 2016 trend is.*
Our study builds upon the observational study of Boulton et al. (2022) and therefore we use the same statistical method as they did.

It is of course interesting to test other methods. For example we noted the recent publication by Wang et al. (2023) uses the Theil-Sen estimator. This method is used to fit a line to data with an underlying trend, but it contains some outliers that may contaminate a normal fit. The figure below shows an application of this method to the MPI-ESM1.2-LR ensemble and the observations. Indeed we see somewhat larger trends, in particular for the observational record. The larger trend in observations is probably achieved because the method filters out data at the ends of the observational record. Observations now have the largest positive trend, but there is still a segment of the historical experiment with a larger in magnitude negative trend. Therefore we anticipate that running more experiments will eventually yield a run with a larger positive trend. Furthermore, it is questionable whether the ends of the observational data record should be treated as outliers, or whether they are part of a slow natural fluctuation.

[Figure]

*Figure 1: Theil-Sen slopes fitted to the AR(1) trends of observations and MPI-ESM1-2-LR for historical and control runs. TS denotes the slope of the fitted line, color scheme accordingly to Fig. 1 in the manuscript.*

**Specific comments**

- *l. 13: After reading the manuscript, this strong statement does not seem to be supported by the presented evidence. Where in the manuscript is the role of local actors studied?*
  Changes made to line 13.

- *l. 57: Can you give references or examples for increasing AR(1) due to other physical reasons?*
  Reference added to line 57.

- *Sect. 2.1: The description of the used simulations is very short. More information on the complexity of the employed models and similarities/differences between them would be very helpful. In particular, do they all simulate dynamic vegetation (variable PFT coverage frequencies)? If not, are there systematic differences between models with and without dynamic vegetation?*
  See comment above.

- *l. 78: Can you expand on why VOD and LAI are physically closely related?*
  See comment below.

- *l. 78-79: While the manuscript states that LAI and VOD are strongly correlated, this seems to be not the case in the Amazon catchment area (Sect. 4.4 and Fig. 11 in Moesinger et al., 2020). This absence of a strong positive correlation in the Amazon rain forest should be stated explicitly and the use of LAI in models and VOD in observations should be justified in light of this weak correlation.*
  Moesinger et al. (2020) demonstrates that VOD reflects plausible seasonal and short-term changes in vegetation and vegetation dynamics on top of LAI and other related optical biophysical vegetation products from optical remote sensing. A positive correlation is shown globally and possible reasons for negative correlation in the Amazon are given (e.g. drought events that increase LAI and decrease VOD).
  This issue is more likely due to a problem with the observational products, in a model the LAI and VOD are to the best of knowledge tightly related in parameterisations, and also physically we see no obvious reason why the opposite should be the case. However, VOD is not provided as output, hence we use LAI.

- *l. 104-105: This sentence might be misleading. To my understanding, the employed Kolmogorov-Smirnov test does not explicitly test for similarity of the spatial structure (i.e., spatial correlations) but only compares the distribution of the tau values across all considered grid boxes. For example, if all the tau values would be randomly reshuffled (thereby loosing the spatial information), it would not change the result of the KS test.*
  Changes made to line 104.

- *l. 119: One 'together' too much*
  Changes made to line 119.

- *l. 137-143: What are the implications of the fact that the KS-test is only passed for so few ensemble members?*
  As explained later in the manuscript, it implies that the observations are on the edge of what models are able to reproduce. In case of higher simulated mean

tau, we would see much more members that pass the KS-test, indicating that a resilience loss is more probable according to models. Changes made to lines 142 – 143.

- *l. 149-150: It is not clear to me if such a strong differentiation between IPSL and models, that pass the test for 1 or 2 members, is justified. Can you quantify this difference statistically?*
Changes made to line 150.

- *Table 2: Please explain in the caption how pcml  is computed*
Changes made to caption of Tab. 2.

- *Fig. 3: It would be insightful to also plot the KS tests for the other models, for example in a supplement. As a major novelty of the paper is the use of multiple models, it does not seem justified to focus on MPI-ESM in several figures and not give equal exposure to all models (or at least the four models with 20 or more ensemble members)*
The KS-test plots for the models with 20 or more ensemble members are going to be provided in the next version of the manuscript.

- *l. 152: Can you explain how the significance is computed here?*
Changes made to line 153.

- *l. 159-167: Why is only MPI-ESM1-2-LR used here and not the other ESMs with plausible ensemble members? Table 2 suggests to me that MIROC-ES2L should be given a larger role in the analysis compared to MPI-ESM since it possesses the most ensemble members that pass the KS-test under historical forcing.*
We chose MPI-ESM for the analysis of SSP5 because we can link the historical runs to the future scenario runs (highlighted members 10 and 28 in Figure 5). This was unfortunately not possible with e.g. MIROC-ES2L that only provides 10 SSP5 runs.

- *l. 165-166: Is the land use change enforced by the SSP scenario? If yes, it should be easy and insightful to quantify its role.*
Figure A7 added. Changes made to line 166.

- *l. 171 Boulton et al. (2022) instead of Boulton et al. 2020?*
Changes made to line 171.

- *l. 174: "are" instead of "is"*
Changes made to line 174.

- *l. 178: "underperform" instead of "under performs"*
Changes made to line 178.

- *l. 178-179: While similar trends to the observations are found in a few ensemble members, the conclusions should emphasize stronger how rarely they occur in the ensembles. For example, the observation's percentiles are at most 0.02 for all piControl ensembles (Table 2) and at most 0.02 for all historical ensembles except for MIROC-ES2L (which is also the only model with a significant change between PI and historical). While it is true that the results suggest that the trend 'could' occur due to internal variability, the likelihood of this occurrence should be the more important quantity, as it is common in other attribution studies.*

We acknowledge the likelihood of the AR(1) trend occurrence e.g. in line 8, 153 – 154, Fig. 4, and new in changed lines 142 – 143.

- *l. 182-183: Can you expand on why the results suggest that "large scale weather events" are responsible for the anomalies?*
  We provide possible reasons for large in magnitude AR(1) trends in models and their spatial distribution. Changes made to line 183.

- *l. 189-191: The presented results only analyze trends in AR(1), interpreted as an indicator of rain forest resilience. Where in the study are the reasons for the decline in rain forest extent studied and where is the role of local actors analyzed?*
  Changes made to lines 189 – 191.

**References**

*WANG, Huan et al. Anthropogenic disturbance exacerbates resilience loss in the Amazon rainforests. Global Change Biology, v. 30, n. 1, p. e17006, https://doi.org/10.1111/gcb.17006, 2023.*

---

## Author Comment (AC2)

**Response to review comments 2**

We want to express our gratitude for the detailed comments of reviewer 2 on the manuscript "Observation-inferred resilience loss of the Amazon rain forest possibly due to internal climate variability". In this document we will respond and discuss in detail the submitted comments.

*The manuscript is well written, concise and offers a thorough explanation of the objectives, methods and a good overview of the results. The work follows some of the steps taken by Boulton et al (2022), using lag-1 year auto-correlation (AR(1)) of changes in forest vegetation optical depth (VOD) and Leaf Area Index (LAI) as an early warning signal before reaching critical threshold for forest dieback.*

*In the results, the authors remark that it is impossible, based solely a single data record, to figure out whether observed changes and trends in auto-correlation of VOD are caused by external forcing or internal variability. To address this problem, they mention that common approaches to detect external (forced) changes are the use of multiple climate model runs, based on different codes, or to use models that start from different initial conditions. They then show, as an example, 30 simulations of the historical experiment, comparing it to one CMIP6 model (MPI-ESM1-2-LR). They show that the observed spatially averaged trend of AR(1) falls within the range of trends of the ensemble members for the same period. They also mention the possibility of investigating simulations of equal length as the historical data, but modeled for a pre-industrial context. After mentioning both examples, they state that the forced response of the Amazon rain forest is not needed to generate an AR(1) increase of similar magnitude as the observed one. And just then they mention that not all models are equally fit for the purpose of simulating the dynamics of the forest in accordance with observations. They then apply the KS test to verify if each ensemble member could have been drawn from the same underlying distribution as the observations, and only four of the nine models pass the test. The model used in both examples mentioned above is one that passes the KS test (MPI-ESM1-2-LR), as it should be. But presenting the above-mentioned examples first, with an important argument for the conclusion, and then showing the "screening" made by using the KS test, may cause some confusion. Thus, I suggest the KS test result is presented before the examples are shown, for increased clarity of the text.*

Whereas we can see the reviewers point of the presentation order can leave some readers concerned, we think the approach taken here is easier to comprehend for the uninitiated and hence more pedagogical. We will therefore keep the order of our presentation.

*Also, after the second example, the authors state (line 125) "The shorter-term deviations appear slightly muted in this case compared to the historical ensemble, and even if encapsulating the longer term trend the shorter variations are clearly less than those observed, therefore possibly suggesting an influence from historical forcing. Overall, however, the displayed model ensemble suggests that the forced response of the Amazon rain forest is not needed to generate an increase in AR(1) of similar magnitude to that observed". In other words, two arguments that support an influence from historical forcing are shown, and then a generic argument is given to argue that forced response is not important. Since the main conclusion of the manuscript is that external forcing do not generate a significant change in AR(1), I think a more detailed discussion to support this at this point would be interesting.*

Changes made to lines 126 – 127. We note that the short term deviations of the

control simulation appear slightly muted compared to the histrocial experiment. However, when comparing the magnitude of the maximum trends (maximum tau in control = 0.79 and in historical = 0.73) and the overall trend ranges, the AR(1) series suggest that the forced response in LAI resilience is rather marginal or slightly negative in MPI-ESM1-2-LR. This is supported by Fig. 4 later in the manuscript. Moreover, the largest trend is found among the control runs, hence the occurrence of the observational trend is more likely in this experiment.

*In the conclusions, the authors state: "Of the four well performing model ensembles, three of them showed trends similar to observations also in their unforced control simulations. These results suggest that the observed trend could simply be an expression of internal variability, and that longer data records would be needed to show that the opposite is the case." Is it possible to give an educated guess on how much longer should the data records be?*

One could calculate the trend distribution over different time scales from the control simulations, and in principle it should become narrower with time. However, we do not have longer observational records available so we do not know the trend that we will observe on longer time scales.

*Boulton et al (2022) argue that the Amazon is showing signs of resilience loss during a period with three "one-in-a-century" droughts, and the higher frequency of extreme droughts leads to ecological changes, but the replacement of drought sensitive tree species by drought resistant ones happens in a slower pace, which may reduce forest resilience even further. If data was available, could the inclusion of the latest extreme drought (2023) to adjust the models significantly change the outcome of the analysis of this manuscript?*

The extension of the data record is highly desirable, as increasing drought risk e.g. in the Eastern Amazon is predicted for the future (Duffy et al., 2015). It is therefore possible that the outcome of the manuscript changes with more data being available.

*The authors also conclude that "This result is further corroborated by the spatial distribution of the increasing trend in AR(1) in the model simulations. Here it is found that ensemble members with substantial positive or negative trends show these in relatively large regions, but not necessarily in those regions with large anthropogenic deforestation. This suggests that such anomalies are associated with large scale weather events." Boulton et al. (2022) compare AR(1) in pixels that are in 50 km bands of distance to areas with human activities to determine the importance of the internal forcings in the resilience loss. Would it be possible to make a similar approach with the modeled data, comparing AR(1) auto-correlation values on a pixel basis, in fixed distance bands to impacted areas? Another possibility is the approach used by Wang et al. (2023), who used forest degradation and deterioration maps to test the human impact on auto-correlation values.*

Due to the low resolution of the GCMs of effectively 100 km – 500 km, an approach like this does not seem very promising in our case.

*Below are a few suggestions of corrections:*

- o *Line 58: "…prior critical transitions,…" should read "… prior to critical transitions,…*
  Changes made to line 58.

o *Legend Table 2: I suggest the authors include the explanation of "piControl" in the legend of the table*
Changes made to caption of Table 2.

o *Correct citation: Boulton et al (2022) is incorrectly cited as Boulton et al. 2020 in line 171.*
Changes made to line 171.

o *In lines 176/177 the phrase "and one model with 33 realisations did clearly under performs." should read "underperform".*
Changes made to line 176 – 177.

**References**

*Boulton, C. A., Lenton, T. M., and Boers, N.: Pronounced loss of Amazon rainforest resilience since the early 2000s, Nature Climate Change, 12, 271–278, https://doi.org/10.1038/s41558-022-01287-8, 2022.*

*WANG, Huan et al. Anthropogenic disturbance exacerbates resilience loss in the Amazon rainforests. Global Change Biology, v. 30, n. 1, p. e17006, https://doi.org/10.1111/gcb.17006, 2023.*

*Philip B. Duffy, Paulo Brando, Gregory P. Asner, and Christopher B. Field: Projections of future meteorological drought and wet periods in the Amazon, Proceedings of the National Academy of Sciences, 43, 112, 13172-13177, doi:10.1073/pnas.1421010112, 2015.*

*Citation*: https://doi.org/10.5194/egusphere-2023-2734-RC2